# Red-Skin Extracts of Lotus Seeds Alleviate High-Fat-Diet Induced Obesity via Regulating Lipoprotein Lipase Activity

**DOI:** 10.3390/foods11142085

**Published:** 2022-07-13

**Authors:** Hong Xu, Hang Gao, Feiyue Liu, Lingxiao Gong

**Affiliations:** 1China-Canada Joint Laboratory of Food Nutrition and Health (Beijing), Beijing Engineering and Technology Research Center of Food Additives, School of Food and Health, Beijing Technology & Business University (BTBU), Beijing 100048, China; xuhong@th.btbu.edu.cn (H.X.); lfyydysa@163.com (F.L.); 2Beijing Academy of Food Sciences, Beijing 100068, China; gaohang0928@163.com

**Keywords:** lotus, polyphenols, obesity, lipoprotein lipase

## Abstract

In recent years, obesity has become an epidemic and an important public health concern. This study was designed to explore the anti-obesity effects of red-skin extracts (RSE) from lotus seeds on high-fat-diet (HFD)-fed mice. In this study, a total of 55 phenolic compounds from the RSE were tentatively characterized using a UPLC-Q/TOF-MS system, including 9 phenolic acids and derivatives, 40 flavonoids, 2 proanthocyanidin, and 4 coumarins and derivatives. Our data demonstrated that RSE could significantly ameliorate obesity characteristics of HFD-fed mice by regulating tissue specific lipoprotein lipase (LPL) activities. In detailed, the activity and expression of LPL in adipose tissue was inhibited, and the activity and expression of LPL in skeletal muscle tissue was enhanced. Overall, these findings suggested that RSE from the red skin of lotus seeds could serve as a great candidate for a value-added, functional ingredient due to its anti-obesity effects via the regulation of LPL activity.

## 1. Introduction

Obesity is often associated with chronic diseases such as arteriosclerosis, hypertension, and diabetes [1,2,3]. It has now become one of the most important health issues of modern society around the world. The occurrence of obesity is due to the energy imbalance between calorie intake and calorie expenditure [4]. Controlling food intake and increasing energy expenditure are the two important principles during the weight-loss process. Lipoprotein lipase (LPL) plays an important role in catalyzing the hydrolysis of the triglycerides-rich lipoproteins, chylomicrons, and very low-density lipoproteins into fatty acids and monoacylglycerol, thereby changing lipids metabolism [5]. These fatty acids and monoacylglycerol are either used by the muscles for metabolic energy or re-esterified into triglyceride (TG) and stored as lipids in adipose tissue. Therefore, if skeletal muscle-specific LPL activity is increased, or adipose tissue-specific LPL activity is decreased, it may be able to reduce fat accumulation and increase energy expenditure, achieving the purpose of anti-obesity. Consequently, compounds that can regulate activity of LPL in muscles tissue and adipose tissue are supposed to function as anti-obesity agents. Recently, it has been proven that polyphenols could regulate the activity of LPL and then ameliorate obesity and its associated ailments. Huang et al. [6] found that green tea polyphenols notably up-regulated LPL activity in skeletal muscles to alleviate obesity in broiler chickens. Liu et al. [7] suggested that polyphenol-enriched extract of *Rosa rugosa* thunb. could significantly increase LPL activity by induction of AMPK signaling activity. Serisier et al. [8] reported that dogs who were given green-tea extract for 12 weeks could significantly down-regulate LPL expression in adipose tissues and up-regulate LPL expression in skeletal muscle compared with baseline.

Lotus (*Nelumbo nucifera* Gaertn.) is an important flowering aquatic plant that has been cultivated for over 3000 years in China [9]. Today, lotus is widely cultivated and honored in Asia, America, and Oceania [10]. The Chinese herb “Lian-Zi”, which is the seed of the lotus, has been used for medicinal purposes as an astringent, tonic, and sedative in oriental countries [11]. Lotus seed red skin is the seed coat of lotus seed, which is the inedible part and thus the by-product of lotus seed manufacturing. According to statistics, the production of red skin from lotus seed reaches about 5000 tons per year in China, resulting in the waste of resources and environmental pollution [12]. Nowadays, red skin from lotus seed is attracting more and more attention due to its healthy properties [13]. Red skin from lotus seed contains abundant functional components, including polysaccharide, polyphenols, and alkaloids, which are associated with anti-oxidation, anti-obesity, anti-inflammatory, and other pharmacological activities. Kredy et al. [14] proved that red skin of lotus seed contained high level of glycosylated flavonols, which exhibited beneficial disease prevention due to the high antioxidant properties. Red skin from lotus seed has been reported to possess the abundance of polyphenols, thereby supporting anti-obesity effects. Cao et al. [15] found that the phenol extract of lotus seed red skin administration significantly lowered the serum leptin level and improved the serum and liver lipid profiles, suppressing lipid peroxidation in hepatic tissue. Further, the extract ameliorated obesity, insulin resistance, and oxidative damage in obese mice. However, there is no other detailed information about the anti-obesity property of the red skin from lotus seed, especially via regulating lipoprotein lipase activity. Moreover, the interaction mechanism between lotus seed red skin and anti-obesity has not been clarified.

Therefore, this study was designed to explore the anti-obesity effects of RSE from lotus seed on high-fat diet-fed mice and determine the anti-obesity properties through regulating the LPL activity in vivo. Furthermore, the interaction mechanism between RSE and LPL were identified to explore the action of regulating LPL activity. The obtained information may help in elucidating the novel intervention effect of RSE from lotus seeds in high-fat-diet-induced obesity and seeking a potential compound with excellent anti-obesity attributes.

## 2. Materials and Methods

### 2.1. Materials and Chemicals

Red skin of lotus seeds was purchased from Sanxianglianye Trading Co., Ltd. (Hangzhou, China) Polyamide was purchased from Changfeng Chemical Co., Ltd. (Changzhou, China) All the other chemicals used were either of analytical grade or of the highest purity commercially available.

### 2.2. Preparation of Lotus Seed Red-Skin Extracts

A portion of 10 g sample powder was dissolved in 180 mL of 60% ethanol overnight at 49 °C. The mixture was ultrasonicated for 9.5 min at 30 °C at 300 W. The ethanol extracts of red skin were then concentrated in a rotary evaporator at 40 °C [16]. The concentrated solution was purified on a polyamide resin column. The column was stepwise eluted successively with deionized water and 90% ethanol solution. The organic phase was collected and freeze-dried as the red-skin extracts [2]. The extracts were stored at −20 °C for chemical composition and biological activity analysis.

### 2.3. Identification of Phenolic Compounds by UPLC-TOF-MS

The UPLC-TOF/MS analysis was performed using a Nexera LC-30A (Shimadzu) coupled to the Triple TOF^TM^ 5600+ (AB SCIEX^TM^, Singapore), a Hybrid Quadrupole-TOF LC/MS/MS Mass Spectrometer, with electrospray ionization (ESI) in positive and negative modes. The separation was performed on a UHPLC column (150 mm × 2.1 mm, 1.7 μm, Waters BEH C18) at a flow rate of 0.3 mL/min. The mobile phase composed of an association of solvent A (water with 0.1% formic acid) and solvent B (acetonitrile). The initial composition of B was 5% and increased to 20% at 4 min, 95% at 12 min, and was then maintained at 95% to 16 min.

For MS analysis, in an information-dependent acquisition (IDA) model, the optimized mass condition was as follows: positive ion source temperature 500 °C, negative ion source temperature 450 °C, ion source gas 1 (GAS 1) 50 psi, ion source gas 2 (GAS 2) 50 psi, Curtain gas (CUR) 25 psi, and ion spray voltage floating (ISVF) 5500 V (positive) and 4400 V (negative). The TOF mass scan range was *m*/*z* 100–1200 Da with an accumulation time of 0.2 s, and the product ion scan range was 50–1000 Da with an accumulation time of 0.01 s. SWATH MS/MS experiments acquired spectra in high-sensitivity mode (declustering potential (DP) ± 60 V, collision energy 35 ± 15 eV).

### 2.4. Animals Study

Ninety male Kunming mice (6 weeks old, body eight 20.0 ± 2.0 g) were purchased from Beijing Vital River Laboratory Animal Technology Co., Ltd. (Beijing, China) All the animal experiments were allowed and performed according to the Guidelines of the Laboratory Animal Ethical Committee of Beijing Vital River Laboratory Animal Technology Co., Ltd. (P2020033). The mice were kept in a regular environment room; the temperature was maintained at 20 ± 2 °C; the humidity was 40–60%; and a light and dark cycle of 12 h was implemented in specific, pathogen-free conditions. After an adaptation period of 1 week, mice were weighed and randomly divided into three treatment groups (*n* = 30 per group). The normal chow group (NC group) was fed with normal chow diet (D12450J: 10 kcal% fat, 20 kcal% protein, 70% kcal carbohydrate, Beijing vital river laboratory animal technology Co., Ltd. (Beijing, China) with supplementary 200 μL water by intragastric gavage. The high-fat-diet group (HFD group) was fed with a high-fat diet (D12492: 60 kcal% fat, 20 kcal% protein, 20% kcal carbohydrate, Beijing vital river laboratory animal technology Co., Ltd.) (Beijing, China) with administration of 200 μL water by intragastric gavage. The RSE treatment group (RSE group) had HFD with treatments of 500 mg/kg·bw/d red-skin extracts by intragastric gavage. All the mice had ad libitum access to diet and water. During the period of experiment, the food consumption and body weight were recorded weekly. At weeks 3, 6, and 9, 30 mice (including 10 mice from NC group, 10 mice from HFD group, and 10 mice from RSE group) were anesthetized with diethyl ether after overnight fasting for 12 h, and blood samples were collected using vacuum tubes containing sodium citrate. The plasma was isolated by centrifugation under 3000 r/min at 4 °C for 10 min and stored at −80 °C for further analysis. Epididymis adipose tissues and skeletal muscle tissues were immediately frozen in liquid nitrogen and stored at −80 °C for further studies.

### 2.5. Serum Biochemical Analyses

Serum levels of total cholesterol (TC), triglyceride (TG), low-density lipoprotein cholesterol (LDL-C), and high-density lipoprotein cholesterol (HDL-C) were analyzed using commercial kits (Nanjing Jiancheng Bioengineering Institute, Nanjing, China).

### 2.6. LPL Activity Measurements

Frozen epididymis adipose tissues and skeletal muscle tissues were homogenized in 9 volumes of physiological saline. The homogenates were centrifuged for 20 min at 3100× *g*, 4 °C. Aliquots of the supernatants were used for determination of tissue LPL activity according to the direction of commercial kits (Nanjing Jiancheng Bioengineering Institute, Nanjing, China). Activity of LPL was expressed in μmol fatty acid liberated per (hour mg protein).

### 2.7. Western Blot Analysis

Western blot analysis was performed according to a previously reported literature with some slight modifications [17]. Briefly, epididymis adipose tissues and skeletal muscle tissue samples were homogenized and lysed in radio immunoprecipitation assay (RIPA) buffer. Then, the lysates were centrifuged at 14,000× *g* for 20 min at 4 °C to obtain the supernatant. The protein content of the supernatant was quantified by using a protein assay kit. The protein concentration for each sample was adjusted to an equal amount with different volumes of loading buffer and denatured in boiling water for 5 min. Then, equal amounts (20 μg) of proteins were loaded onto 10% SDS-polyacrylamide gel electrophoresis and then transferred to polyvinyl difluoride membranes under the condition of 1.5 A and 20 V for 60 min. The membranes were blocked with 5% albumin bovine serum containing 0.1% Tween-20 (TBST) for 60 min and washed with 0.1% Tris Buffered Saline Tween incubated (TBST) for three times followed by incubation overnight at 4 °C with primary antibodies. After that, the membranes were washed with 0.1% TBST for 15 min and incubated with secondary antibodies for another 1.5 h. Finally, membranes were visualized with ECL kit and scanned by ChemiDoc XRS^+^. β-actin was used as the internal control.

### 2.8. RNA Extraction and RT-PCR Analysis

Epididymis adipose tissues and skeletal muscle tissues samples were cut into small pieces and homogenized. The total RNA was extracted with the TRIzol reagent following the manufacturer’s protocol and quantified using a Nanodrop 1000 spectrophotometer. First-strand cDNA synthesis was performed using an iScript reverse transcriptase kit. Quantitative real-time polymerase chain reaction (qRT-PCR) was conducted using the SYBR Green mix and ABI 7900 HT. qRT-PCR conditions were operated by the procedure as described in commercial introduction. Statistical analysis of qRT-PCR was performed using the 2^−ΔΔCT^ method, which calculated the relative changes in gene expression of the target normalized to endogenous reference gene GAPDH [18]. Specific forward and reverse primer sequences as follows were used in this study: LPL (Forward: 5′- GCCTGAGTTGTAGAAAGAATCG -3′, Reverse: 5′- CTCAGTCCCAGAAAAGTGAATC-3′); GAPDH (Forward: 5′- ACAGCAACAGGGTGGTGGAC -3′, Reverse: 5′-TTTGAGGGTGCAGCGAACTT-3′).

### 2.9. Statistical Analysis

Data are reported as the mean ± standard deviation (SD). Statistical significance was assessed by the one-way ANOVA procedure followed by Tukey test using SPSS software (Version 22 for Windows, IBM, New York, NY, USA). Statistical significance was declared at *p* < 0.05 or *p* < 0.01.

## 3. Results and Discussions

### 3.1. Identification or Tentatively Characterization of Phenolic Compounds in RSE

A total of 55 phenolic compounds from the RSE were identified or tentatively characterized using a UPLC-Q/TOF-MS system, including 9 phenolic acids and derivatives, 40 flavonoids, 2 proanthocyanidin, and 4 coumarins and derivatives (Table 1). Base peak intensity chromatograms in positive and negative mode of RSE are shown in Figure 1. The phenolic compounds were tentatively identified by analyzing and comparing their retention times, observed *m*/*z*, MS spectra, major MS^2^ fragment ions, as well as the MS data in the *MassBank, Respect*, and *GNPS* databases.

The detected phenolic acid compounds included 2-acetylacteoside (**1**), 1-(3,4-dimethoxycinnamoyl) piperidine (**3**), (E)-3-[2-[ (2S,3R,4S,5S,6R) -3,4,5- trihydroxy-6- (hydroxymethyl) oxan-2-yl] oxyphenyl] prop-2-enoic acid (**7**), 2,5-dihydroxybenzoic acid (**10**), protocatechuic aldehyde (**18**), methyl rosmarinate (**19**), isoferulic acid (**38**), sinapic acid (**42**), taxifolin (**46**), and sinapoylcholine (**54**). Among them, components **1**, **7**, **19**, and **55** were coumaric acid derivatives, component **3** was a cinnamic acid derivative, and component **45** was a sinapic acid derivative. The monomers easily produced fragment ions by eliminating small molecules, such as H_2_O (18 Da), CO (28 Da), and CO_2_ (43Da).

The 41 flavonoids tentatively identified in RSE could be further divided into flavone aglycones, C-glycosyl flavonoids, and O-glycosyl flavonoids. The identified flavone aglycones included gallocatechin (**6**), casticin (**11**), 5,7-dihydroxy-2-(3-hydroxy-4-methoxyphenyl)-2,3-dihydro-4H-chromen-4-one (**15**), catechin (**17**), (–)-epigallocatechin (**21**), 7-hydroxy-3-phenyl-4H-chromen-4-one (**25**), quercetin (**27**), 8-(azepan-1-ium-1-ylmethyl)-3- (2,4-dimethoxyphenyl) -2-oxo-2H-chromen-7-olate (**30**), 4′,7-Di-O-methylnaringenin (**31**), 2-(3,4-dihydroxyphenyl)-5-hydroxy-10-isopropyl-9,10-dihydropyrano [2,3-f] chromene-4,8-dione (**33**), 7-hydroxy-3-phenyl-4H-chromen-4-on (**36**), isoquercitin (**39**), kaempferol (**43**), 4′,5,7-Trihydroxy-6,8-diprenylisoflavone (**44**), isorhamnetin (**48**), genistein (**51**), and diosmetin (**52**).

Overall, there were 13 O-glycosyl flavonoids, including 2 isoflavonoid derivates (**14**, **45**), 1 myricetin derivate (**23**), 3 quercetin derivates (**28**, **29**, **40**), 1 isorhamnetin derivate (**49**), 1 vitexin derivate (**53**), 9-(2,3-dihydroxypropoxy)-9-oxononanoic acid (**50**), spiraeoside (**32**), liquiritin (**34**), hyperoside (**41**), and 3-[(2S,3R,4S,5R,6R)-3,5-dihydroxy-6- (hydroxymethyl)-4- [(2S,3R,4R,5R,6S)-3,4,5- trihydroxy-6-methyloxan-2-yl]oxyoxan-2-yl]oxy-5,7-dihydroxy-2-(4-hydroxyphenyl)chromen-4-one (**47**). The O-glycosyl flavonoids showed high intensity of aglycone via losing saccharide groups, such as rhamnose (146 Da), C_6_H_10_O_5_ (162 Da), and rhamnosylglucose (308Da). Components **2**, **4**, **20**, **22**, **24**, **35**, and **37** were all categorized as C-glucosides flavonoids.

Besides flavonoids, there were four coumarins and derivatives, including 6-methylcoumarin (**5**), 6-methoxy-4-methyl-2H-chromen-2-one (**8**), praeruptorin A (**13**), and 3,4-dihydrocoumarin (**55**); two retrocharlcones, including chalcone (**9**) and 2-hydroxychalcone (**26**); and two proanthocyanidins, including procyanidin B2 (**12**) and procyanidin C1 (**16**).

### 3.2. RSE Attenuated the Features of Obesity and Dyslipidaemia in HFD-Fed Mice

As shown in Figure 2a, the impact of RSE on the above obesity parameters was not attributed to reduced energy intake but primarily due to RSE intervention since no significant difference was found in energy intake between all the groups (Figure 2a). As shown in Figure 2b, there were no significant differences in the initial body weight of mice among the three groups. However, after 3 weeks of intervention, mice fed with HFD had higher body weight (Figure 2b) and epididymal fat index (Table 2) and developed hallmark features of dyslipidemia, including elevated TC (Figure 2c), TG (Figure 2d), and LDL-c (Figure 2e) levels. The HDL-c levels had no differences among the groups. Intervention with RSE significantly attenuated HFD-induced overweight, epididymal fat index, and serum TC, TG, and LDL-c levels of obese mice (*p* < 0.01).

### 3.3. RSE Regulated the LPL Activity in HFD-Fed Mice

The effects of RSE on the LPL activity in epididymal adipose and skeletal muscle were shown in Figure 3. Compared with NC group, the LPL activity of epididymal adipose in the HFD group was elevated 77.96%, whereas the LPL activity in skeletal muscle tissue was markedly depressed 49.44% (of the control value) at 9 weeks. After 9 weeks of RSE intervention, the LPL activity of epididymal adipose in the obese mice was markedly depressed by 90.90% compared to HFD group. In contrast, the LPL activity of skeletal muscle in the RSE-treated mice significantly increased by 1.22 times as compared with HFD group. Our study suggested that the LPL activity could be modulated by RSE to regulate the state of obesity.

### 3.4. RSE Regulated the LPL Protein Expression in HFD-Fed Mice

Figure 4 indicates the effect of RSE interventions on LPL protein expression in different tissues, respectively. Notably, LPL protein level increased significantly in adipose tissue in response to HFD treatment, while it was reduced in skeletal muscle under HFD (*p* < 0.01). Additionally, the LPL protein expression showed time-dependency in adipose or skeletal muscle tissues in HFD group. As expected, after 9 weeks of RSE treatment, the LPL protein level markedly decreased in adipose tissue (*p* < 0.01, Figure 4a) and significantly increased in skeletal muscle (*p* < 0.01, Figure 4b).

### 3.5. RSE Regulated the LPL mRNA Expression in HFD-Fed Mice

As shown in Figure 5, LPL mRNA levels were found to be significantly increased in epididymal adipose tissue, whereas it significantly decreased in skeletal muscle after HFD treatment for 6 weeks (*p* < 0.01). The intervention of RSE significantly reduced the LPL mRNA expression in the epididymal adipose of mice fed HFD (*p* < 0.01). Besides, RSE treatment increased the LPL mRNA expression in skeletal muscle by 4.3-fold as compared with the HFD group.

## 4. Discussions

Obesity, a worldwide epidemic nowadays, is a metabolic disease associated with high blood pressure and diabetes [1]. The imbalance between calorie intake and calorie expenditure is highly related to the occurrence of obesity [19]. It has been reported that the LPL plays a vital role in energy expenditure and lipid metabolism. LPL is a multifunctional enzyme produced in various tissues, especially adipose and skeletal muscle tissue, where it plays different roles. Skeletal muscle contains at least half the total body LPL, and the free fatty acids generated by muscle LPL are catabolized for energy. Therefore, muscle, not adipose tissue, was the primary site of chylomicron triglyceride fatty acid clearance and energy balance [20]. Accumulating evidence suggests that the decrease of LPL activity in epididymal adipose and the improvement in skeletal muscle has the beneficial effect of preventing diet-induced obesity [21]. Herein, we investigated whether the anti-obesity effect of RSE is due to LPL activity regulation. In this study, RSE intervention significantly attenuated increased body weight and obesity characteristics, including serum TC, TG, and LDL-c levels. Lotus-leaf extract is reported to prevent body weight increase and reduce the TC, TG, and LDL-c levels and increase the HDL-c levels in the high-fat-diet-induced obese mice [22]. However, few reports to date have elucidated the effect of red-skin extract on obesity.

Notably, a remarkable decreased activity of LPL was detected in epididymal adipose, and a significant increased activity of LPL was detected in skeletal muscle after intervention with RSE for 9 weeks. Additionally, the intervention of RSE significantly reduced the LPL mRNA expression in the epididymal adipose but increased the LPL mRNA expression in skeletal muscle. The results of this study demonstrated that RSE mediates LPL expression in mice with tissue-specific differences.

As reported, the increase in adipose tissue weight and lipid contents were concomitant with the increase in LPL activities involved in the lipid storage [23]. LPL overexpression in muscle attenuates weight gain by potentiating energy expenditure [24]. In agreement with our findings, Del Bas et al. reported that LPL mRNA levels are decreased in adipose tissue and increased in muscle after administration of grape seed procyanidins in rats [25]. Moreover, extract of fermented *Curcuma longa* L., which is rich in curcuminoids, prevented obesity by inducing a statistically significant decrease in the mRNA expressions of LPL in the white adipose tissue of obese mice [26]. Similar results were for the effect the activity of LPL in the skeletal muscle of polysaccharide from *Auricularia auricula* in cholesterol-enriched-diet-fed mice [27]. Additionally, extracts of lotus leaves, abundant with flavonoids (kaempferitrin, hyperoside, and astragalin), significantly upregulated mRNA expression in the livers of obese mice [22]. These findings of our study indicate that red-skin extract of lotus may partially inhibit adipogenesis by controlling fatty acid uptake into adipose tissue from circulating lipoproteins. The results also suggested that RSE prevented obesity through preferentially directing triglycerides to energy production by the muscle instead of to energy storage by the adipose tissue via the regulation of LPL activity in different tissues. As demonstrated, the LPL expression is regulated by several upstream tissue-specific regulatory factors, such as uncoupling proteins (UCPs), adenosine monophosphate (AMP)-activated protein kinase phosphorylation (pAMPK), and angiopoietin-like proteins (ANGPTLs). These regulators in adipose and muscle tissues were reported to have tissue-specific responses to dietary interventions. For example, the LPL activator NO-1886 increased the UCP 3 expression and LPL activity only in skeletal muscle [28]. AMPK activation or its phosphorylation resulted in the decreases of LPL activity in adipocytes and the increase of LPL activity in skeletal muscle tissue [29]. Further studies should be undertaken to better understand the molecular mechanisms of these opposing regulation effects of RSE on LPL activity in different tissues.

It has been shown that phenolic compounds may influence the lipoprotein composition by regulating LPL, which is involved in the dynamic exchange of lipids between the different lipoproteins in tissues [30]. Phenolic compounds comprise an array of substances such as procyanidins, rutin, vitexin, vitexin-glucoside, hyperoside, and quercetin, and may have direct or indirect effect of the regulation of LPL [25,31,32]. However, this effect is not clear because no changes are observed in LPL mRNA levels of muscle and adipose tissue after the administration of chronic grape seed procyanidins to hyperlipidemic rats [33]. Moreover, hawthorn flavonoids, with a purity of 93.8%, comprised of vitexin, vitexin-rhamnoside, vitexin-glucoside, rutin, hyperoside, pinnatifin, and quercetin, mediated LPL expression in mice with tissue-specific differences by which LPL increased significantly in muscular tissues and decreased in adipose tissues [30].

Phenolic compounds, especially flavonoids, have been widely accepted as a category of the most important bioactive compounds of lotus. The phenolic compounds in lotus-derived parts were mainly identified as phenolic acids, flavonoids, and their glycoside derivatives [9]. Among all the tentatively identified phenolic compounds in this study, catechin, gallocatechin, quercetin, quercetin-3-glucuronide, isoquercetin, kaempferol, and rutin were reported in lotus leaf, flower, embryo, or/and stamen [34]. Of note, luteolin glucoside, hyperin, kaempferol glycoside, quercetin glycoside, diosmetin 7-O-hexose, and isorhamnetin 3-O-arabino-pyranosyl-(1,2)-glucopyranoside were also identified in lotus-leaf extract [35]. For lotus seed skins, procyanidin B1, (+)-catechin, procyanidin B4, procyanidin B2, and (+)-gallocatechin were identified. And these isolated procyanidin dimers exhibited in vivo antioxidant activities via the activation of nuclear factor-E2-related factor 2 (Nrf2)–antioxidant response element (ARE) signaling pathway [12]. Most of the phenolic compounds in the red skin of lotus seed, including phenolic acids, flavonoids, proanthocyanidin, coumarins, and derivatives, were first identified in this study. Moreover, among these compounds, vitexin, vitexin-glucoside, rutin, hyperoside, and quercetin were reported to modulate the LPL activity with tissue-specific way. Therefore, the LPL modulation effect of RSE may directly relate to the phenolic compounds. Further study should be conducted to identify the components of RSE that affect LPL activity and its regulators.

## 5. Conclusions

In conclusion, our results revealed that red-skin extract of lotus seeds, which is composed of 55 phenolic compounds, could alleviate body weight, adipose tissue weight, and serum TC, TG, and LDL-c in high-fat-diet-induced obese mice. These results can be associated with the promotion of energy expenditure in the muscle and suppression energy storage in the adipose tissue via the regulation of LPL activity. The present study demonstrated that red-skin extract of lotus seed has great potential as a functional food in the prevention of obesity-related diseases.

## Figures and Tables

**Figure 1 foods-11-02085-f001:**
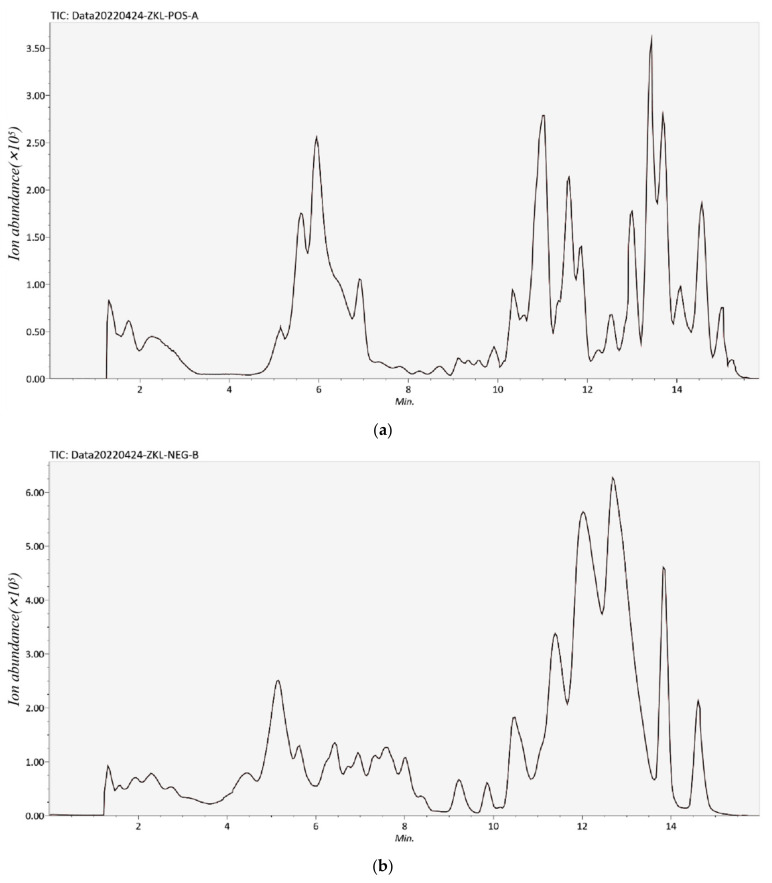
Base peak intestine chromatograms of RSE in positive ion mode (**a**) and negative ion mode (**b**).

**Figure 2 foods-11-02085-f002:**
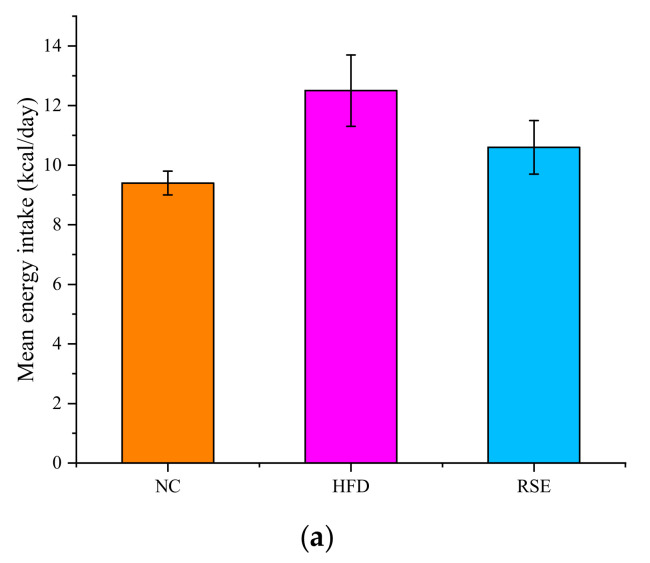
RSE prevented high-fat-diet-induced obesity. (**a**) Mean energy intake per mouse per day. (**b**) Body weight changes during 9-week intervention. (**c**) Serum total cholesterol (TC) level. (**d**) Serum triglyceride (TG) level. (**e**) Serum low-density lipoprotein cholesterol (LDL-C) level. (**f**) Serum high-density lipoprotein cholesterol (HDL-C) level. Data are expressed as the mean ± SEM. ^#^
*p* < 0.05 and ^##^
*p* < 0.01 for NC vs. HFD; * *p* < 0.05 and ** *p* < 0.01 for RSE vs. HFD. NC, the normal chow group; HFD, the high-fat-diet group; RSE, the RSE treatment group.

**Figure 3 foods-11-02085-f003:**
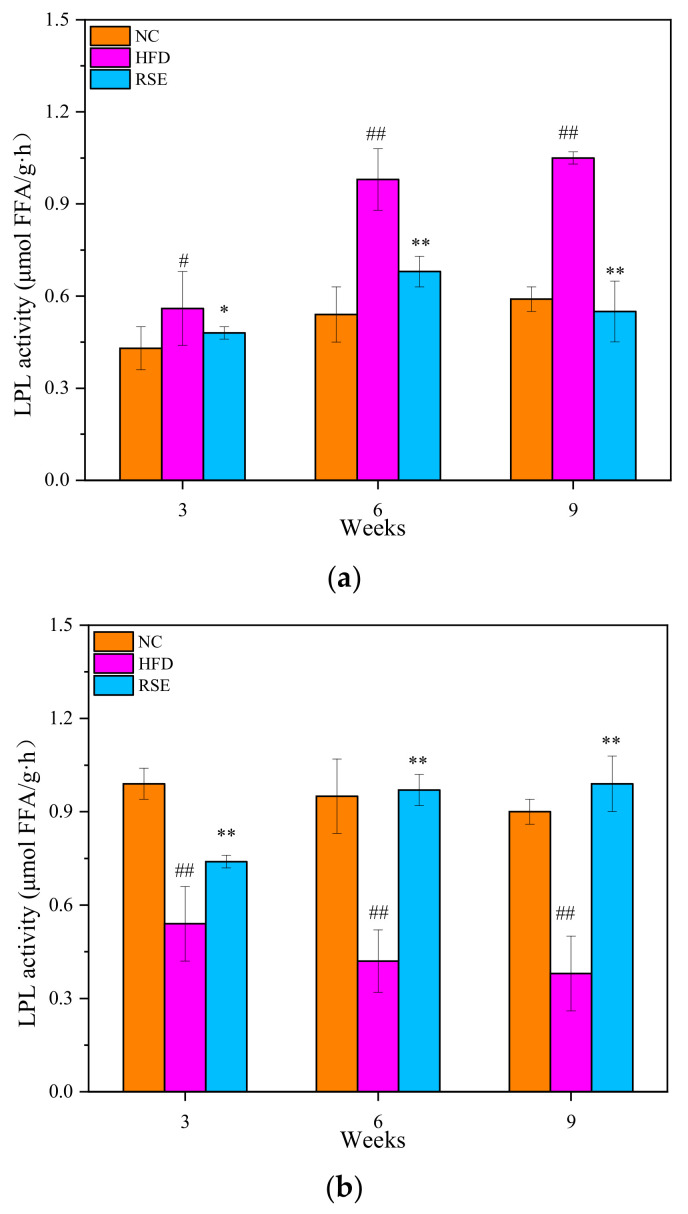
Effect of RSE on the LPL activity in epididymal adipose (**a**) and skeletal muscle (**b**). Data are expressed as the mean ± SEM. ^#^
*p* < 0.05 and ^##^
*p* < 0.01 for NC p. HFD; * *p* < 0.05 and ** *p* < 0.01 for RSE vs. HFD. Group abbreviations: refer to Figure 2 caption.

**Figure 4 foods-11-02085-f004:**
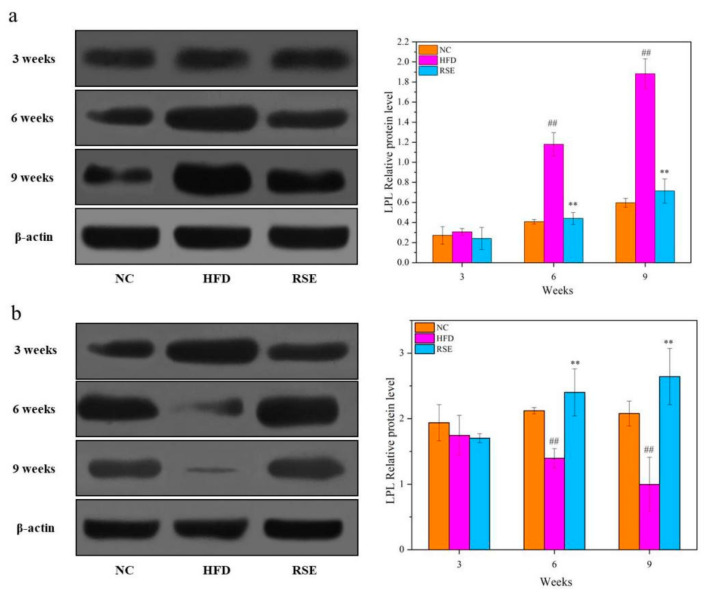
Effects of RSE on the LPL relative protein level in epididymal adipose tissue (**a**) and skeletal muscle (**b**). Data are expressed as the mean ± SEM. ^##^
*p* < 0.01 for NC vs. HFD; ** *p* < 0.01 for RSE vs. HFD. Group abbreviations: refer to Figure 2 caption.

**Figure 5 foods-11-02085-f005:**
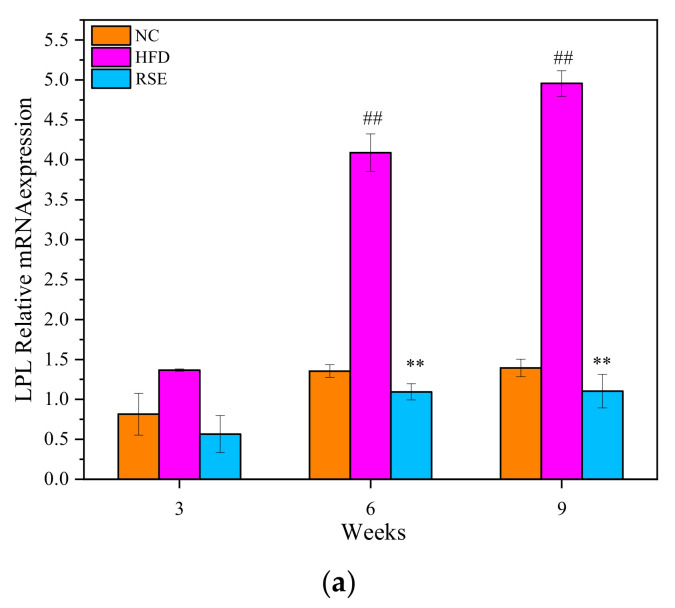
Effects of RSE on the LPL relative mRNA expression in epididymal adipose tissue (**a**) and skeletal muscle (**b**). Data are expressed as the mean ± SEM. ^#^
*p* < 0.05 and ^##^
*p* < 0.01 for NC vs. HFD; * *p* < 0.05 and ** *p* < 0.01 for RSE vs. HFD. Group abbreviations: refer to Figure 2 caption.

**Table 1 foods-11-02085-t001:** The information of compounds identified in RSE.

No.	Proposed Compound	RT (min)	Observed *m*/*z*	Adduct	Theoretical *m*/*z*	Formula	Ontology	Major Fragment Ions *m*/*z*
1	2-Acetylacteoside	1.315	689.2020	[M+Na]+	689.2000	C31H38O16	Coumaric acids and derivatives	527, 525, 509, 467, 365, 347
2	Luteolin-6-C-glucoside	1.433	447.0968	[M-H]−	447.0933	C21H20O11	Flavonoid C-glycosides	401, 357, 327, 297, 285
3	1-(3,4-Dimethoxycinnamoyl) piperidine	1.434	298.1414	[M+Na]+	298.1400	C16H21NO3	Cinnamic acids and derivatives	283, 282, 254, 240, 190, 107
4	Apigenin 6-C-glucoside 8-C-arabinoside	1.472	563.1463	[M-H]−	563.1406	C26H28O14	Flavone C and C-glycosides	473, 443, 383, 353
5	6-Methylcoumarin	1.593	161.0593	[M+H]+	161.0600	C10H8O2	Coumarins and derivatives	143, 115
6	(+)-Gallocatechin	1.671	305.0669	[M-H]−	305.0667	C15H14O7	Flavanol	125
7	(E)-3-[2-[(2S,3R,4S,5S,6R)-3,4,5-trihydroxy-6-(hydroxymethyl) oxan-2-yl] oxyphenyl] prop-2-enoic acid	1.671	371.1010	[M+FA-H]−	371.0980	C15H18O8	Coumaric acids and derivatives	163, 119
8	6-methoxy-4-methyl-2H-chromen-2-one	1.673	191.0861	[M+H]+	191.0800	C11H10O3	Coumarins and derivatives	189
9	Chalcone	1.713	209.0936	[M+H]+	209.0961	C15H12O	Retrochalcones	189
10	2,5-dihydroxybenzoic acid	1.751	153.0198	[M-H]−	153.0193	C7H6O4	Hydroxybenzoic acid derivatives	152, 124, 109, 107
11	Casticin	1.832	397.0908	[M+Na]+	397.0900	C19H18O8	7-O-methylated flavonoids	283
12	Procyanidin B2	1.909	577.1404	[M-H]−	577.1351	C30H26O12	Biflavonoids and polyflavonoids	289, 245, 205, 125
13	Praeruptorin A	1.912	409.0904	[M+Na]+	409.0890	C21H22O7	Angular pyranocoumarins	325
14	6″-O-Acetylglycitin	1.912	511.1231	[M+Na]+	511.1200	C24H24O11	Isoflavonoid O-glycosides	475, 379
15	5,7-dihydroxy-2-(3-hydroxy-4-methoxyphenyl)-2,3-dihydro-4H-chromen-4-one	1.951	325.0706	[M+Na]+	325.0700	C16H14O6	4′-O-methylated flavonoids	325
16	Procyanidin C1	1.989	865.2050	[M-H]−	865.198	C45H38O18	Biflavonoids and polyflavonoids	739, 695, 577, 407, 287
17	Catechin	2.276	289.0721	[M-H]−	289.0718	C15H14O6	Flavanol	245, 203, 151, 125, 109
18	Protocatechuic aldehyde	2.752	137.0244	[M-H]−	137.0244	C7H6O3	Hydroxybenzaldehydes	136, 119, 109, 108
19	Methyl rosmarinate	4.943	397.0910	[M+Na]+	397.0900	C19H18O8	Coumaric acids and derivatives	379, 349, 307, 295
20	Swertisin	4.943	469.1074	[M+Na]+	469.1100	C22H22O10	Flavonoid C-glycosides	433, 375
21	(-)-Epigallocatechin	5.024	329.0643	[M+Na]+	329.0640	C15H14O7	Flavonal	311, 300, 286, 283, 255, 149
22	Isoshaftoside	5.064	587.1323	[M+Na]+	587.1300	C26H28O14	Flavonoid 8-C-glycosides	569, 551, 509, 497
23	Myricetin-3-Galactoside	5.237	479.0872	[M-H]−	479.0831	C21H20O13	Flavonoid-3-O-glycosides	433, 316, 271
24	Homoorientin	5.316	447.0942	[M-H]−	447.0933	C21H20O11	Flavonoid C-glycosides	369, 357,327, 299, 298, 297, 285
25	7-hydroxy-3-phenyl-4H-chromen-4-one	5.353	239.0678	[M+H]+	239.0700	C15H10O3	Isoflavones	224, 181
26	2-Hydroxychalcone	5.393	225.0904	[M+H]+	225.0910	C15H12O2	Retrochalcones	197, 182
27	Quercetin	5.513	303.0482	[M+H]+	303.0500	C15H10O7	Flavonols	285, 274, 257, 247, 229, 201, 153, 137
28	Rutin	5.513	633.1376	[M+Na]+	633.1405	C27H30O16	Flavonols O-glycoside	331
29	Quercetin-3-O-rutinoside	5.593	609.1476	[M-H]-	609.1461	C27H30O16	Flavonoid O-glycosides	343, 301, 300, 271, 255, 242, 178
30	8-(azepan-1-ium-1-ylmethyl)-3-(2,4-dimethoxyphenyl)-2-oxo-2H-chromen-7-olate	5.593	432.1784	[M+Na]+	432.1800	C24H27NO5	Hydroxyisoflavonoids	297, 174
31	4′,7-Di-O-methylnaringenin	5.633	323.0882	[M+Na]+	323.0900	C17H16O5	7-O-methylated flavonoids	295
32	Spiraeoside	5.633	465.1124	[M+H]+	465.1030	C21H20O12	Flavonoid O-glycosides	303
33	2-(3,4-dihydroxyphenyl)-5-hydroxy-10-isopropyl-9,10-dihydropyrano [2,3-f] chromene-4,8-dione	5.644	405.0939	[M+Na]+	405.0900	C21H18O7	Pyranoflavonoids	387, 377
34	liquiritin	5.644	441.1163	[M+Na]+	441.1200	C21H22O9	Flavonoid O-glycosides	423, 405, 379, 351, 325
35	Vitexin	5.671	431.1003	[M-H]−	431.0984	C21H20O10	Flavonoid 8-C-glycosides	341, 323, 311, 283
36	7-hydroxy-3-phenyl-4H-chromen-4-one	5.683	239.0677	[M+H]+	239.0700	C15H10O3	Isoflavones	211, 183, 165, 153
37	5,7-dihydroxy-2-(4-hydroxyphenyl)-6-[(2S,3R,4R,5S,6R)-3,4,5-trihydroxy-6-(hydroxymethyl)oxan-2-yl]-8-[(2S,3R,4R,5R,6S)-3,4,5-trihydroxy-6-methyloxan-2-yl]chromen-4-one	5.710	577.1602	[M-H]−	577.1570	C27H30O14	Flavonoid 8-C-glycosides	487, 473, 457, 383, 353, 179
38	3-Hydroxy-4-methoxycinnamic acid (isoferulic acid)	5.750	193.0518	[M-H]−	193.0506	C10H10O4	Hydroxycinnamic acids	178, 134
39	Isoquercitin	5.750	463.0902	[M-H]−	463.0880	C21H20O12	Flavonoid-3-O-glycosides	301, 300, 271, 255
40	Quercetin-3-Glucuronide	5.750	477.0706	[M-H]−	477.0675	C21H18O13	Flavonoid-3-O-glucuronides	301, 255, 179, 151
41	Hyperoside	5.750	927.1992	[2M-H]−	927.1837	C21H20O12	Flavonoid-3-O-glycosides	463, 301
42	Sinapic acid	5.789	223.0645	[M-H]−	223.0612	C11H12O5	Hydroxycinnamic acids	193, 149
43	Kaempferol	5.803	287.0529	[M+H]+	287.0550	C15H10O6	Flavonols	287, 269, 241, 161, 153, 135
44	4′,5,7-Trihydroxy-6,8-diprenylisoflavone	5.803	429.1610	[M+Na]+	429.1700	C25H26O5	6-prenylated isoflavanones	309
45	(2R,3R,4S,5R,6R)-2-(acetoxymethyl)-6-((3-(benzo[d][1,3]dioxol-5-yl)-6-ethyl-4-oxo-4H-chromen-7-yl)oxy)tetrahydro-2H-pyran-3,4,5-triyl triacetate	5.962	663.1678	[M+Na]+	663.1600	C32H32O14	Isoflavonoid O-glycosides	417
46	taxifolin	5.868	303.0550	[M-H]−	303.0510	C15H12O7	Sinapic acid derivative	285, 177, 125
47	3-[(2S,3R,4S,5R,6R)-3,5-dihydroxy-6-(hydroxymethyl)-4-[(2S,3R,4R,5R,6S)-3,4,5-trihydroxy-6-methyloxan-2-yl]oxyoxan-2-yl]oxy-5,7-dihydroxy-2-(4-hydroxyphenyl)chromen-4-one	5.868	593.1551	[M-H]−	593.1512	C27H30O15	Flavanone glycosides	547, 285, 255
48	Isorhamnetin	5.962	317.0674	[M+H]+	317.0656	C16H12O7	Flavonol	302, 285, 274, 245, 229, 153
49	Isorhamnetin-3-O-rutinoside	6.027	623.1674	[M-H]−	623.1617	C28H32O16	Flavonol O-glycosides	315, 314, 300
50	9-(2,3-dihydroxypropoxy)-9-oxononanoic acid	6.187	261.1359	[M-H]−	261.1348	C12H22O6	Flavonol O-glycosides	187, 125
51	Genistein	6.121	271.0629	[M+H]+	271.0601	C15H10O5	Isoflavones	153
52	Diosmetin	6.200	301.0710	[M+H]+	301.0700	C16H12O6	4′-O-methylated flavonoids	286, 258, 229
53	4′-O-Glucosylvitexin	6.680	617.1353	[M+Na]+	617.1400	C27H30O15	Flavonoid O-glycosides	555
54	Sinapoylcholine	7.320	310.1599	[M]+	310.1643	C16H24NO5	Coumaric acids and derivatives	278, 253
55	3,4-Dihydrocoumarin	11.650	149.0583	[M+H]+	149.0597	C9H8O2	3,4-dihydrocoumarins	121, 103

**Table 2 foods-11-02085-t002:** Epididymal fat index during the intervention periods.

Group	3 Weeks	6 Weeks	9 Weeks
NC	4.76 ± 1.134	11.61 ± 2.84	12.26 ± 3.67
HFD	24.61 ± 9.49 ^##^	24.35 ± 7.67 ^##^	29.30 ± 8.95 ^##^
RSE	14.35 ± 4.42 ^##^**	17.38 ± 5.35 ^##^**	19.67 ± 3.07 ^##^**

Data are expressed as the mean ± SEM. ^##^
*p* < 0.01 as compared with NC group; ** *p* < 0.01 as compared with HFD group. The epididymal fat index is expressed as mg/g. Group abbreviations refer to Figure 2 caption.

## Data Availability

Data are contained within the article.

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
