# Peer review of "Red-Skin Extracts of Lotus Seeds Alleviate High-Fat-Diet Induced Obesity via Regulating Lipoprotein Lipase Activity"

_foods, 2022, doi:10.3390/foods11142085_

Round 1

Reviewer 1 Report

Interesting research, please add the discussion about the active compound or active compound group content in the extract used.

The purposes is also related to the active compound, but I could not find in conclusion.

Other comment please see in the manuscripts.

Author Response

Dear Editors and Reviewers,

We do appreciate for Editors and Reviewers’ warm work earnestly. Thank you for your valuable comments concerning our manuscript entitled “Red-Skin extracts of lotus seeds alleviate high fat diet induced obesity via regulating lipoprotein lipase activity” (foods-1808120). Those comments are all valuable and very helpful for revising and improving our paper, as well as the important guiding significance to our research. We have carefully revised the manuscript point-by-point according to your comments, and the revised parts are highlighted in the revised manuscript.

We hope that the changes made adequately address the concerns raised, and the detailed responses are listed as below.

Response to Reviewer #1:

  1. Please explain LPL first.

Response:

The full name of LPL was added in abstract as following:

  Our data demonstrated that RSE could significantly ameliorate obesity characteristics of HFD-fed mice by regulating tissue specific lipoprotein lipase (LPL) activities.

  1. 60 is ethanol or water?

Response:

The statement of “60% aqueous ethanol” was corrected as “60% ethanol”.

  1. Please change rpm into g.

Response:

The statement of “5000 rpm” was corrected as “3100 g”.

  1. Please add the note for NC, HFD, and RSE.

Response:

According to reviewer’s advice, the note for NC, HFD, and RSE were added in Figure 2 caption as following:

Figure 2. RSE prevented high fat diet-induced obesity. (a) Mean energy intake per mouse per day. (b) Body weight changes during 9-week intervention. (c) Serum total cholesterol (TC) level. (d) Serum triglyceride (TG) level. (e) Serum low-density lipoprotein cholesterol (LDL-C) level. (f) Se-rum high-density lipoprotein cholesterol (HDL-C) level. Data are expressed as the mean ± SEM. #p < 0.05 and ##p < 0.01 for NC vs HFD; *p < 0.05 and **p < 0.01 for RSE vs HFD. NC, the normal chow group. HFD, the high fat diet group. RSE, the RSE treatment group.

  1. Please add the “and”.

Response:

The statement of “Obesity, a worldwide epidemic nowadays, is a metabolic disease associated with high blood pressure, diabetes” was corrected as “Obesity, a worldwide epidemic nowadays, is a metabolic disease associated with high blood pressure and diabetes”.

  1. No conclusion about specific compound.

Response:

This study was designed to explore the anti-obesity effects of red skin extract (RSE) from lotus seed on high-fat diet-fed mice. Thus, the specific composition in RSE was identified (including 9 phenolic acids and derivatives, 40 flavonoids, 2 proanthocyanidin and 4 coumarins and derivatives). Those compounds might play anti-obesity effects collectively rather than a specific compound.

Reviewer 2 Report

This is an interesting manuscript examining potential anti-obesity responses to a food waste product. 

The red skin extract of lotus seeds decreased LPL in adipose yet increased LPL in skeletal muscle, similar to literature studies, for example with green tea polyphenols. The molecular mechanisms of these opposing effects should be examined in the Discussion.  

Are these changes in LPL the only or even the most important mechanism for the anti-obesity activity? Other mechanisms such as anti-oxidant responses and prevention of inflammatory cell infiltration are also likely, based on literature studies. How could the relative importance of the different mechanisms be established?

The following publication has not been cited but may provide evidence for a different molecular mechanism of action:

Lotus seed skin proanthocyanidin extract exhibits potent antioxidant property via activation of the Nrf2–ARE pathway Tao Li,  Qili Li,  Weiguo Wu,  Yong Li,  De-xing Hou,  Hua Xu,  Baodong Zheng, Shaoxiao Zeng,  Yang Shan,  Xiangyang Lu ... Acta Biochimica et Biophysica Sinica, Volume 51, Issue 1, December 2019, Pages 31–40, https://doi.org/10.1093/abbs/gmy148

Which flavonoid is the most likely to have this biological activity on LPL regulation? The authors should suggest studies with this compound alone as future research.

Since flavonoids such as rutin are very cheap and readily available, what is the commercial or therapeutic advantage of the extraction of the red lotus skin?

The authors use a prevention protocol rather than a reversal protocol. In humans, most anti-obesity treatments are aimed at already obese patients, so treatment is usually a reversal protocol. Does a prevention protocol necessarily indicate that existing obesity can be reversed?

Table 1: Please add the concentrations of defined compounds so that daily dose can be estimated. The dose of a drug is critical in defining the likely therapeutic response. The doses in rats can then be used to estimate effective human doses using the Reagan-Shaw formulae. 

Please include the identities of compounds with RT (retention times) above 8 minutes, other than dihydrocoumarin, from Figure 1 in Table 1.

Future research should also include characterisation of cardiovascular responses as cardiovascular disease is a major complication of obesity. Further, histological studies of fat deposition in the liver and skeletal muscle would greatly enhance the power of this study so should be undertaken.

Minor points:

Lines 45-50: add review on health benefits of lotus seed?

Lines 49-50 – reference for production

Lines 112, 114: Source of pre-made diets and composition

Lines 119-121: cut-and-paste mistake

Figure 2a: y-axis legend should be kcal/day

Prefer figures 2a and c-f to be in Table so that other researchers can see the values.

Author Response

Dear Editors and Reviewers,

We do appreciate for Editors and Reviewers’ warm work earnestly. Thank you for your valuable comments concerning our manuscript entitled “Red-Skin extracts of lotus seeds alleviate high fat diet induced obesity via regulating lipoprotein lipase activity” (foods-1808120). Those comments are all valuable and very helpful for revising and improving our paper, as well as the important guiding significance to our research. We have carefully revised the manuscript point-by-point according to your comments, and the revised parts are highlighted in the revised manuscript.

We hope that the changes made adequately address the concerns raised, and the detailed responses are listed as below.

Response to Reviewer #2:

  1. The red skin extract of lotus seeds decreased LPL in adipose yet increased LPL in skeletal muscle, similar to literature studies, for example with green tea polyphenols. The molecular mechanisms of these opposing effects should be examined in the Discussion.  

Response:

According to the comments, the molecular mechanisms of the opposing effects were discussed in the revised manuscript as following:

As demonstrated, the LPL expression is regulated by several upstream regulators such as uncoupling proteins (UCPs), adenosine monophosphate (AMP)-activated protein kinase phosphorylation (pAMPK). These regulators in adipose and muscle tissues were reported to have tissue-specific responses to dietary interventions. For example, the LPL activator NO-1886 increased the UCP 3 expression and LPL activity only in skeletal muscle. AMPK activation or its phosphorylation resulted in the decreases of LPL activity in adipocytes, while increases of LPL activity in skeletal muscle tissue. Further studies should be taken to clarify the molecular mechanisms of these opposing regulation effects of RSE on LPL activity in different tissues.

  1. Are these changes in LPL the only or even the most important mechanism for the anti-obesity activity? Other mechanisms such as anti-oxidant responses and prevention of inflammatory cell infiltration are also likely, based on literature studies. How could the relative importance of the different mechanisms be established?

Response:

Thank you very much for your comments. They are very helpful for our future work. LPL is the key enzyme responsible for hydrolysis of triacylglycerol in circulating lipoproteins and chylomicrons to release free fatty acids for tissue storage or utilization. The LPL protein level and activity are tightly regulated by multiple mechanisms. For example, hypoxia inducible factor 1 (HIF-1α), tumor necrosis factor-α (TNF-α) can alter the protein level and activity of LPL,thereby changing lipid deposition in tissues and related diseases/complications (Clinica Chimica Acta, 2020, 503,19-34). HIF-1α is the transcription factor that involved in the redox regulation. And TNF-α plays an important role in inflammation. Therefore, anti-oxidant and anti-inflammatory responses might be the upstream regulatory factor for the subsequent changes in LPL activity and expression. In order to better understand the relevance of three mechanisms, the molecular mechanisms of the regulation effects of RSE on LPL activity should be further studied.   

  1. The following publication has not been cited but may provide evidence for a different molecular mechanism of action:

Lotus seed skin proanthocyanidin extract exhibits potent antioxidant property via activation of the Nrf2–ARE pathway Tao Li,  Qili Li,  Weiguo Wu,  Yong Li,  De-xing Hou,  Hua Xu,  Baodong Zheng, Shaoxiao Zeng,  Yang Shan,  Xiangyang Lu ... Acta Biochimica et Biophysica Sinica, Volume 51, Issue 1, December 2019, Pages 31–40, https://doi.org/10.1093/abbs/gmy148

Response:

According to the comments, the molecular mechanisms of the lotus seed skin proanthocyanidin for antioxidant activity is stated in the revised manuscript as following:

For lotus seed skins, procyanidin B1, (+)-catechin, procyanidin B4, procyanidin B2 and (+)-gallocatechin were identified. And these isolated procyanidin dimers exhibited in vivo antioxidant activities via the activation of nuclear factor-E2 related factor 2 (Nrf2) – antioxidant response element (ARE) signaling pathway

  1. Which flavonoid is the most likely to have this biological activity on LPL regulation? The authors should suggest studies with this compound alone as future research.

Response:

Thank you very much for your comments. They are very helpful for our future work. Among these identified phenolic compounds in red skin extracts, vitexin, vitexin-glucoside, rutin, hyperoside and quercetin were reported to modulate the LPL activity with tissue-specific way. The reported studies have been suggested in the manuscript for the future research.  

  1. Since flavonoids such as rutin are very cheap and readily available, what is the commercial or therapeutic advantage of the extraction of the red lotus skin?

Response:

   In our study, the anti-obesity effect of red-skin extracts form lotus seeds instead of certain compounds were demonstrated in mice. It is suggested that red skin extracts of lotus seeds could serve as a great candidate for value-added functional ingredient due to anti-obesity effects via the regulation of LPL activity. Our further study will be conducted to identify the bioactive components with anti-obesity effects.   

  1. The authors use a prevention protocol rather than a reversal protocol. In humans, most anti-obesity treatments are aimed at already obese patients, so treatment is usually a reversal protocol. Does a prevention protocol necessarily indicate that existing obesity can be reversed?

Response:

       Both the prevention protocol and reversal protocol were widely used to screen the potential functional ingredients for the prevention of obesity-related diseases (Foods. 2021, 10(6), 1266. https://doi.org/10.3390/foods10061266) (J Food Biochem. 2022;00:e14200. https://doi.org/10.1111/jfbc.1420). Our results revealed that red skin extract of lotus seeds, which composed of 55 phenolic compounds could alleviate body weight, adipose tissue weight, serum TC, TG and LDL-c in high-fat diet-induced obese mice. The results also suggested that red skin extracts prevented obesity through preferentially directing triglycerides to energy production by the muscle instead of to energy storage by the adipose tissue via the regulation of LPL activity in different tissues. It is implied that the regulation effect of red skin extracts on LPL may also be happened in the obese mice, thereby alleviating the obesity-related characteristics. In order to demonstrate this outcomes, more studies should be conducted in the future.    

  1. Table 1: Please add the concentrations of defined compounds so that daily dose can be estimated. The dose of a drug is critical in defining the likely therapeutic response. The doses in rats can then be used to estimate effective human doses using the Reagan-Shaw formulae. 

Response:

Thank you very much for your comments. They are very helpful for our future work. In this study, the phenolic compounds are tentatively characterized by UPLC-TOF-MS. The concentrations of the compounds were not determined. The anti-obesity effect of red-skin extracts form lotus seeds instead of certain compounds were demonstrated in mice. The dosage of the extract was 500 mg/kg per day. According to the Reagan-Shaw formulae, the effective doses of red-skin extract for human is 40.5 mg/kg per day. Our further study will be conducted to identify the components of red-skin extract that affect the LPL activity and its regulators. And the concentration of defined compounds will be further studied.

  1. Please include the identities of compounds with RT (retention times) above 8 minutes, other than dihydrocoumarin, from Figure 1 in Table 1.

Response:

As identified by the UPLC-TOF-MS, the compounds with retention times above 8 minutes are hydrophobic compounds including lignans, neolignans, triterpenoids and related compounds. Our further study will be conducted to elucidate the components of red-skin extract that affect the LPL activity. 

  1. Future research should also include characterisation of cardiovascular responses as cardiovascular disease is a major complication of obesity. Further, histological studies of fat deposition in the liver and skeletal muscle would greatly enhance the power of this study so should be undertaken.

Response:

Thank you very much for your comments. They are very helpful for our future work. The findings of this study suggested that RSE from red skin of lotus seeds could serve as a great candidate for alleviation of obesity in association with the regulation of LPL activity. Further studies should be taken to clarify the anti-obesity effect of red skin of lotus and underlying mechanisms. According to reviewers’ comments, the cardiovascular responses and ft deposition in the liver and skeletal muscle will be taken into account.   

  1. Lines 45-50: add review on health benefits of lotus seed?

Response:

According to reviewer’s advice, the review on health benefits of lotus seed was added in introduction as following:

The Chinese herb ‘‘Lian-Zi’’, which is the seed of the lotus, has been used for medicinal purposes as an astringent, tonic, and sedative in oriental countries.

  1. Lines 49-50 – reference for production.

Response:

The reference for production was added as following:

Zhu M, Liu T, Zhang C, et al. Flavonoids of Lotus (Nelumbo nucifera) Seed Embryos and Their Antioxidant Potential[J]. Journal of Food Science, 2017, 82(7-9):1834-1841.

  1. Lines 112, 114: Source of pre-made diets and composition.

Response:

The source of pre-made diets and composition was added as following:

D12450J: 10 kcal% fat, 20 kcal% protein, 70% kcal carbohydrate, Beijing vital river laboratory animal technology Co., Ltd.

D12492: 60 kcal% fat, 20 kcal% protein, 20% kcal carbohydrate. Beijing vital river laboratory animal technology Co., Ltd.

  1. Lines 119-121: cut-and-paste mistake.

Response:

The statement of “During the period of experiment, the food consumption and body weight were recorded weekly. At the week of 3, 6, and 9, 30 mice (including 10 mice from NC group, 10 mice from HFD group and 10 mice from RSE group) were anesthetized with diethyl ethincluding 10 mice from NC group, 10 mice from HFD group and 10 mice from 120 RSE group) were anesthetized with diethyl ether after overnight fasting for 12 h” was corrected as “During the period of experiment, the food consumption and body weight were recorded weekly. At the week of 3, 6, and 9, 30 mice (including 10 mice from NC group, 10 mice from HFD group and 10 mice from RSE group) were anesthetized with diethyl ether (including 10 mice from NC group, 10 mice from HFD group and 10 mice from RSE group) were anesthetized with diethyl ether after overnight fasting for 12 h”.

  1. Figure 2a: y-axis legend should be kcal/day.

Response:

The y-axis legend was corrected as kcal/day as following:

  1. Prefer figures 2a and c-f to be in Table so that other researchers can see the values.

 Response:

     During the period of experiment, the obesity-related characteristics recorded at the week of 3, 6, and 9. The changes during the feeding period were obviously displayed in the figures. And the values also could be obtained from the figures. Based on this consideration, we would like to show the results in the figure.
